# A signature-agnostic test for differences between tumor mutation spectra reveals carcinogen and ancestry effects
Samuel F. M. Hart [1], Nicolas Alcala [2], Alison F. Feder [1,3,4,5] ✉ & Kelley Harris [1,3,5] ✉

Despite dozens of tools to identify mutational signatures in cancer samples, there is not an established metric for quantifying whether signature exposures differ significantly between two heterogeneous groups of samples. We demonstrate that a signature-agnostic metric - the aggregate mutation spectrum distance permutation method (AMSD) - can rigorously determine whether mutational exposures differ between groups, a hypothesis that is not directly addressed by signature analysis. First, we reanalyze a study of carcinogen exposure in mice, determining that eleven of twenty tested carcinogens produce significant mutation spectrum shifts. Only three of these carcinogens were previously reported to induce distinct mutational signatures, suggesting that many carcinogens perturb mutagenesis by altering the composition of endogenous signatures. Next, we interrogate whether patient ancestry has a measurable impact on human tumor mutation spectra, finding significant ancestry-associated differences across ten cancer types. Some have been previously reported, such as elevated SBS4 in African lung adenocarcinomas, while some have not to our knowledge been reported, such as elevated SBS17a/b in European esophageal carcinomas. These examples suggest that AMSD is a robust tool for detecting differences among groups of tumors or other mutated samples, complementing descriptive signature deconvolution and enabling the discovery of environmental and genetic influences on mutagenesis.

Mutation rates and biases in mutational sequence context can be influenced by endogenous cellular processes, environmental exposures, and genetic variation. All such mutagenic processes can leave decodable imprints on the relative frequency distribution of different possible mutation types in a genome, termed the mutation spectrum, which can be especially complex in the context of cancers. To interpret this variation, a framework has been introduced for jointly decomposing the mutation spectra of thousands of tumors across dozens of cancer types to identify *mutational signatures*: characteristic mutation type distributions that are thought to reflect underlying mutational processes[1–3].

Over 100 mutational signatures are currently listed in the COSMIC database (Catalogue Of Somatic Mutations In Cancer), with the majority being single base substitution (SBS) signatures, which capture the frequencies of nucleotide changes classified by their 3mer flanking nucleotide context (e.g. CCG > CTG)[4]. It is now common practice when analyzing cancer genome sequencing data to infer which mutational signatures are active in a given sample, often revealing key information about each individual cancer's oncogenesis and evolutionary history.

While mutational signature analysis is effective for identifying *why* two mutation spectra may differ, it is not ideal for testing *whether or not* two spectra differ significantly from one another. For a simple comparison between two tumors, a hypergeometric test[5] can be used to judge whether they have significantly different mutation spectra or mutational signature exposures, but more complex approaches are needed to make sense of exposure distributions in larger datasets that can contain thousands of tumors from patients who differ in various ways that might affect mutagenesis. Mutational signature decomposition offers no formal statistical framework for determining which exposures or characteristics are associated with significant differences in tumor mutation profiles. When within-group heterogeneity and between-group heterogeneity are both present, as illustrated in Fig. 1, it is not obvious how to determine whether between-group heterogeneity significantly exceeds within-group heterogeneity (the

[1]Department of Genome Sciences, University of Washington, Seattle, WA, USA. [2]Computational Cancer Genomics Team, International Agency for Research on Cancer (IARC/WHO), Genomic Epidemiology Branch, Lyon, France. [3]Herbold Computational Biology Program, Fred Hutch Cancer Center, Seattle, WA, USA. [4]Howard Hughes Medical Institute, Seattle, WA, USA. [5]These authors jointly supervised this work: Alison F. Feder, Kelley Harris. ✉e-mail: affeder@uw.edu; harriske@uw.edu

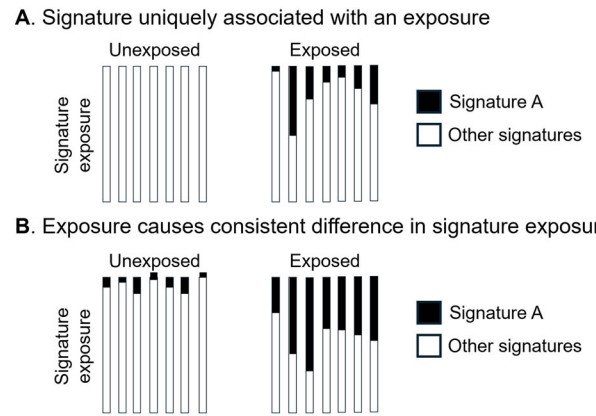

**Fig. 1 | Mutation spectra can differ within and between cohorts in multiple qualitatively different ways.** In each of the above illustrations, there exists variation in mutational exposures both within and between two groups of tumors, one from carcinogen-exposed individuals and one from unexposed individuals. As heterogeneity increases, it becomes more complicated to determine whether the exposure is associated with mutagenic consequences. **A** In this example, Signature A is present in all exposed tumors but no unexposed tumors, making it straightforward to conclude that Signature A is associated with the exposure. **B** Here, Signature A is present in unexposed tumors as well, though at lower relative dosage than seen in any exposed tumors. Although this difference between groups is relatively straightforward to describe, it would not meet the Riva, et al. criteria for associating the chemical exposure with Signature A. **C** If exposure to Signature A is more variable within each group, additional statistical testing is clearly needed to determine whether exposure is higher on average within either group. **D** When multiple mutational signatures are present, it becomes more complex to determine whether an exposure is significantly more prevalent in the exposure group. This complexity is often further exacerbated by differences in the mutation load among samples, which causes signature inference to be more error-prone in samples with fewer mutations.

standard ANOVA test does not apply since there is no straightforward definition of a mutation spectrum's mean or variance).

Comparison of variation within and between groups of tumors is relevant to many important biological questions, such as how carcinogens affect mutagenesis and how mutational processes differ between cancers affecting people from different populations. This limitation is reflected in the results of many papers that study tumor mutagenesis. For example, the effect of carcinogens on mutational signature exposure was the subject of a recent study by Riva et al.[6], but this study restricted its analysis to flagging qualitative differences in the presence or absence of specific signatures and identifying signatures that exclusively appear in the presence of a particular carcinogen exposure (Fig. 1A) but not testing whether any carcinogens appear to alter the intensity of exposure to mutational signatures that are present in both exposed and unexposed tumors (Fig. 1B–D). As cancer sequencing datasets grow larger and gain the resolution to test more hypotheses about what factors influence cancer mutations, the ability to detect quantitative differences in mutational signature exposure will be critical for making the most of the information content of these datasets.

Recently, Sasani et al. developed a metric called the aggregate mutation spectrum distance (AMSD) that measures the differences among the total mutation spectra of a set of samples[7]. This metric was instrumental for mapping genetic variants that affect germline mutagenesis in a recombinant inbred mouse panel, due to its ability to detect relatively small perturbations caused by mutator loci. Although some other studies have applied distance-based approaches to investigate mutation spectrum differences, such as Kucab et al.[8], these studies have generally created bespoke spectrum comparison methods that are not tested for broader generalizability or implemented for easy application to other data sets. Moreover, analyses of tumor mutation spectra usually perform signature decomposition as a first step rather than directly analyzing the signals present in the raw mutation spectra. While this approach can increase the ease of a paper's interpretability, it also propagates any errors and uncertainties associated with mutation spectrum decomposition to all downstream analyses in a way that is not straightforward to measure. Here, we show that AMSD is a powerful tool for testing hypotheses about what variables affect the complex mutation spectra of cancer genomes, generating results that come with $p$-values and in some cases resolving structure that is obscured by traditional signature analysis.

## Results

### AMSD measures differences in signature exposure using a permutation approach

The aggregate mutation spectrum distance test (AMSD) uses a permutation-based metric to measure the statistical significance of mutation spectrum differences between two groups, such as when comparing a carcinogen-exposed cohort to a control cohort (Fig. 2A). AMSD quantifies the difference between the two spectra by calculating a distance metric (e.g. cosine distance), and then uses permutations to randomly reshuffle group labels across samples, recalculating the distance between the mutation spectra of thousands of randomly subsampled groups to generate a null distribution expectation, assuming no true differences between the groups. The fraction of permutations that generate a larger or equal distance than that observed between the real groups can be interpreted as the $p$-value significance of the difference between the two groups' aggregate mutation spectra. Significant differences can then be examined using mutational-signature-based approaches to investigate drivers of mutational divergence (Fig. 2B), which may be biological or artifactual.

Applying AMSD prior to mutational signature analysis has several advantages. Most notably, it provides an unbiased test for whether a statistically significant difference exists prior to interpreting that difference using signatures. In addition, it outputs a single comparison between the groups, reducing output dimensionality and minimizing the multiple testing burden that would be imposed by fitting signatures and then statistically comparing each pair of signature exposures. This is particularly useful when screening many variables for mutagenic effects, which can cause the multiple testing burden to quickly drown out the signal. This multiple testing burden can be reduced by comparing signature exposures only for variables that AMSD flags as significantly affecting spectra. Finally, since null permutations retain the data comparison structure, AMSD controls for unequal sample sizes and mutation counts between groups, a source of noise that is not always easy to see in the context of signature decomposition plots. The shape of the null mutation spectrum distance distribution can also be informative about the structure of the data – for example, a bimodal distribution indicates that spectrum subtypes or outliers are likely driving much of the variation in the data set. AMSD can aggregate spectra with all samples weighted evenly (the default method we use in this paper unless otherwise noted), or with samples weighted by mutation count to capture

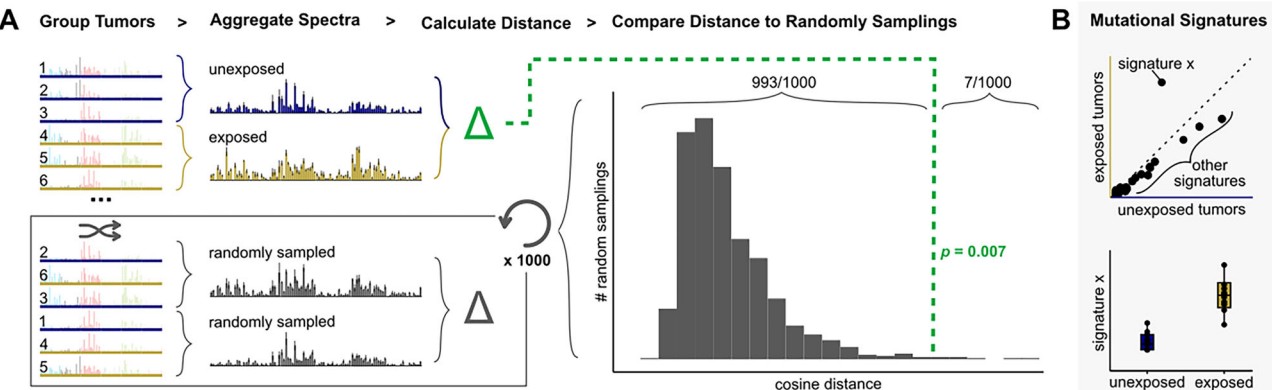

**Fig. 2 | AMSD: a method to detect significant differences between mutation spectra. A** Given mutation spectra for each sample in a cohort divided into two groups, such as tumors unexposed (blue) or exposed (yellow) to a carcinogen, AMSD aggregates mutation spectra for each group and calculates the cosine distance between the aggregate spectra (green). AMSD also randomly reshuffles group labels to calculate the cosine distance between randomly sampled tumors (gray), repeating 1000+ times to create a null distribution expectation. AMSD then calculates a *p*-value from the fraction of random samplings that are greater than or equal to the observed distance between the two groups, as visualized in the histogram. **B** To interpret what mutational mechanisms may underlie a significant difference, mutational signature fitting can be applied to the aggregate spectra (top) or individual samples (bottom) and compared for variables that AMSD finds have a significant effect on mutation spectra.

the effects of highly mutated outliers. For more details on considerations when running AMSD, see methods.

To gain intuition about when AMSD is sufficiently powered to detect mutation spectrum differences, we applied the method to several simulated datasets. We tested AMSD's ability to detect a difference between an exposure group and a control group while varying sample size (i.e., the number of tumors in each comparison group), mutation count, signature "flatness", and signature exposure level. We observed that AMSD can reliably detect a signal when at least two of the following four parameters are met: sample size is large (hundreds), mutation count is high (WGS rather than WES), the variable signature is "spiky" (dominated by a small number of triplet contexts such as SBS2 rather than distributed more uniformly across contexts like SBS40), or relative exposure of the variable signature is high (>10% mutations/sample) (Supplementary Fig. 1A). Although real data sets are likely to be noisier and contain additional confounding variables, these simulations provide a framework to assess when AMSD is likely powered to detect a signal and in which cases a negative result may reflect lack of power rather than true mutational homogeneity.

In theory, the uncertainty associated with mutational signature decomposition guarantees that testing for a difference between the raw mutation spectra of two groups should be less noisy than testing for a difference in signature composition. To empirically test whether this is the case, we compared AMSD's performance to a Wilcoxon rank sum test approach that several studies have used to test individual signatures for differences between groups[9–11]. We extended this Wilcoxon rank sum test approach to test for differences in overall mutational signature composition and applied it to our simulated data as follows: first, we identified exposures of COSMIC signatures within each simulated sample using SigProfilerAssignment[12]. We then used the Wilcoxon rank sum to test each individual signature for an exposure difference between groups, applied a Bonferroni multiple testing correction, and identified the most significant corrected signature *p*-value. This method performed similarly to AMSD across many simulations, but severely underperformed AMSD in cases with i) small sample size, ii) flat signatures, iii) spiky signatures at low mutation counts, and iv) non-COSMIC signature exposure (Supplementary Fig. 1B). The results of this test suggest that performing signature decomposition as a preprocessing step does not increase uncertainty in high-information settings where signature decomposition is very reliable, but that running tests on raw mutation spectra is better practice when data is sparse or signature inference is error-prone for other reasons. It is possible that other tests for differences in signature composition might perform better, as we did not explore the performance of other signature fitting tools, parameters, or

significance methods, each which would incur inherent trade-offs and multiple testing burdens across signatures. It is also worth noting that fewer such choices are necessary when running AMSD to screen for differences in raw mutation spectra, a fact that contributes to our tool's robustness.

## Many carcinogens influence tumor mutation spectra via shifts in relative exposure to endogenous signatures

Environmental exposures to carcinogens such as ultraviolet light and smoking are responsible for some of the most striking and interpretable mutational signatures. However, most carcinogens lack such a striking characteristic signature and have not been reported to have a measurable effect on mutation load or mutational signatures in human cancer databases. To investigate carcinogens in a controlled setting, Riva et al. recently exposed mice to 20 different known or suspected carcinogens and then used whole genome sequencing to read out mutations from the resulting lung and liver tumors, extracting mouse mutational signatures de novo and comparing signature exposures between carcinogen-exposed tumors and spontaneously arising tumors in unexposed mice[6]. Surprisingly, they found that only three of these carcinogens significantly increased mutation load, and only three caused distinct mutational signatures exclusively associated with a mutagen as in Fig. 1A: cobalt, vinylidene chloride (VDC), and 1,2,3-trichloropropane (TCP). However, we noted qualitative differences among the proportions of signature exposures in several other carcinogen-exposed tumors from this dataset. We hypothesized that some mutagens might increase the rates of certain endogenous mutational processes or interfere with their repair, and so we applied AMSD to test which mutagens were associated with significant shifts in the overall mutation spectrum composition (differences resembling the patterns in Fig. 1B–D).

Of 29 mutation spectrum comparisons (liver tumors exposed to one of 20 carcinogens versus spontaneous liver tumors, lung tumors exposed to one of 9 carcinogens versus spontaneous lung tumors), 15 revealed differences between exposed and unexposed tumors at a significance level of $p < 0.05$. 12 of these differences remained significant after Benjamini–Hochberg correction for multiple testing, and 9 remained significant after more stringent Bonferroni correction (Fig. 3A, Supplementary Fig. 2). Three significant differences, including the two most significant comparisons, correspond to the three carcinogens identified in the original study to have specific carcinogen-induced genomic signatures; TCP-liver ($p = 7 \times 10^{-4}$ cosine distance 0.18), VDC-liver ($p = 3 \times 10^{-5}$ cosine distance 0.11) and cobalt-lung ($p < 1 \times 10^{-5}$ cosine distance 0.12), highlighting AMSD's ability to recapitulate signature-based results. There were also many other carcinogens that had no characteristic signature, but appeared

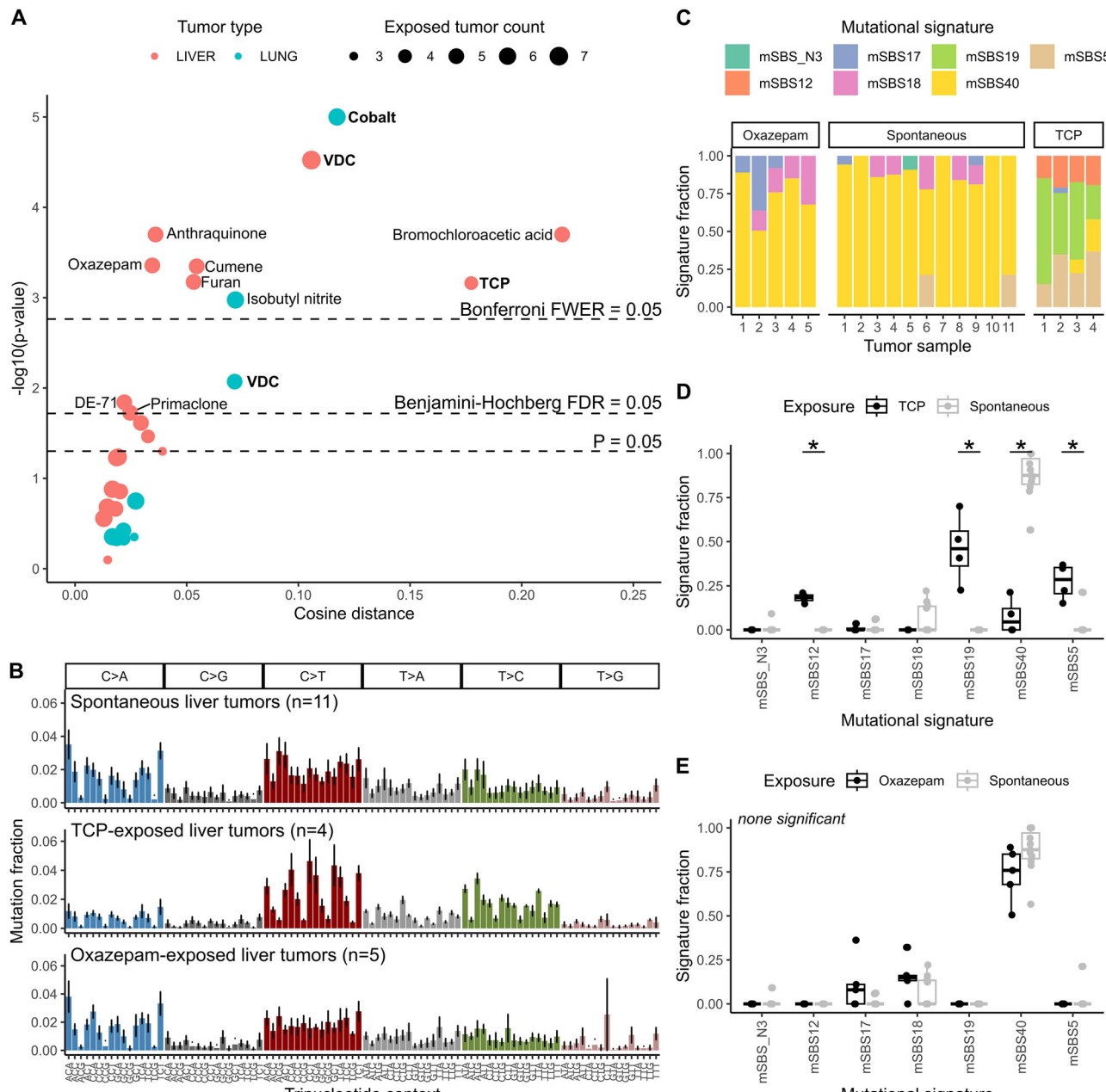

**Fig. 3 | Many carcinogens affect mouse mutation spectra. A** Volcano plot of AMSD results comparing carcinogen-exposed tumors to spontaneous tumors of the same tumor type (liver = red, lung = blue). Point sizes denote the number of carcinogen-exposed tumors, compared to $n = 11$ spontaneous liver tumors or $n = 12$ spontaneous lung tumors. Dashed lines correspond to significance thresholds: individual $P = 0.05$ threshold, Benjamini–Hochberg corrected false discovery rate (FDR) adjusted threshold, and Bonferroni corrected family-wise error rate (FWER) adjusted threshold. Carcinogen-exposed tumors in bold denote those with specific carcinogen-induced mutational signatures reported by Riva et al.[6]. **B** Aggregate mean mutation spectra for spontaneous, TCP-exposed, and oxazepam-exposed liver tumors, with standard deviation in error bars. **C** Stacked bar plots of mutational signature exposure fractions for each sample in (**B**). Mutational signatures were extracted de novo by Riva et al. Side-by-side comparisons for each signature comparing TCP-exposed (**D**) or oxazepam-exposed (**E**) to spontaneous liver tumors. Asterisks denote $P < 0.05$ (both Wilcoxon rank sum test and two-tailed $t$ test with unequal variance were run and were in agreement in all cases).

to cause stark shifts in mutation spectra, as evidenced by high cosine distance (e.g. bromochloroacetic acid liver: $p = 2 \times 10^{-4}$ cosine distance 0.22). AMSD also flagged the significance of some more subtle difference in mutation spectra, as evidenced by low cosine distance (e.g. oxazepam liver: $p = 4 \times 10^{-4}$ cosine distance 0.035).

AMSD reveals which exposures significantly perturb the mutation spectrum, but it does not tell us why spectra differ between exposures. To investigate why spectra differ, it is helpful to compare the spectra themselves and then apply signature fitting (Supplementary Figs. 3, 4). As representative examples, we picked one high cosine distance tumor type that was

flagged by signature analysis, TCP-exposed liver, and one low cosine distance tumor type that was not flagged by signature analysis, oxazepam-exposed liver (Fig. 3B). Visualizing signature exposures for each tumor as stacked bar plots, a common method for displaying signature fitting results, we see TCP tumors have a markedly different signature profile than spontaneous liver tumors, reflecting the signature 12 and 19 exposures that were highlighted in the original study (Fig. 3C, D, Supplementary Fig. 5A). The corresponding human versions of these two signatures do not have a known etiology, although this result in mice indicates that TCP-like compounds may directly or indirectly affect these mutational processes.

For oxazepam, differences in signature exposure fractions are not significantly different for any individual signature (Fig. 3C, E, Supplementary Fig. 5A), but signature decomposition appears to be masking subtle but consistent mutation spectrum differences that exist between oxazepam-exposed and unexposed tumors, such as ACT > ATT mutations (Fig. 3B, Supplementary Fig. 5B). This spectrum difference was present in both intergenic regions ($p = 0.003$, mean mutations/sample = 1166) and genic regions ($p = 0.04$ mean mutations/sample = 618) (Supplementary Fig. 5C). It is possible this difference could be explained by the presence of an unknown signature, or that instead of introducing new mutational processes oxazepam might be shifting tumor cell growth/metabolism or DNA repair in ways that perturb the relative rates of endogenous mutational processes. Ultimately, a larger sample set and complete signature set would be needed to resolve what mutational processes may be driving this difference, but this analysis demonstrates how AMSD and signature fitting can work in concert.

To directly compare a signature-based approach to signature-agnostic AMSD, we applied the Wilcoxon rank sum approach described in the previous section to all 29 comparisons (Supplementary Fig. 5D). AMSD was much more powerful at detecting differences, with the signature-based approach weakened by the combination of multiple testing burden (2–7 signatures for each of 29 comparisons) and low sample sizes ($n \leq 12$). We also compared AMSD to mean within-sample signature diversity, a metric from Morrison et al. that tests whether the sample-to-sample variability in signature presence differs between groups[39], but found only moderate correspondence because the metrics are testing for different signals (Supplementary Fig. 5E, F).

Interestingly, chemicals that induce significant mutation spectrum divergence by AMSD are only nominally more likely to be positive on an Ames test, a bacterial assay used to assess the mutagenic potential of chemical compounds (5/12 versus 3/17, $p = 0.2$, Fisher's exact test). This is consistent with our hypothesis that carcinogens affecting mutation spectra do not necessarily introduce new mutational processes. However, chemicals with significant effects on mutation spectra according to AMSD were more likely to have "clear evidence" for carcinogenicity (11/12) than those without a detectable spectra difference (8/17) ($p = 0.02$, Fisher's exact test) in the results from the National Toxicology Program Bioassay, which uses rodent studies to determine the strength of the evidence that each chemical is carcinogenic. Ultimately, our results indicate that many more carcinogens may affect mutation spectra than previously detected by signature fitting methods, and that AMSD is sensitive to underlying differences in mutation spectra even in experimental data sets with relatively small sample sizes.

One known human carcinogen that has not been linked to a difference in mutational signatures is asbestos, a material used in construction that is known to cause mesothelioma[13–15]. Although they did not find asbestos-related mutational signatures or spectrum changes, Mangiante et al. detected molecular differences that correlate with asbestos exposure, such as the expression of mesenchymal and neoangiogenesis-related genes[15], which could indirectly influence mutation spectra in subtle ways that signature analysis may not be powered to detect. Though we do not detect a difference in SBS spectra between mesothelioma tumors with ($n = 75$) and without ($n = 28$) known professional asbestos exposure in the Mangiante et al. data set ($p = 0.67$), we do detect a significant difference in copy number variant (CNV) spectra ($p = 0.018$) (Supplementary Fig. 6). Similar to SBS, CNV spectra can be generated by classifying CNVs into different types by loss-of-heterozygosity status, total copy number state, and segment length. This asbestos-associated CNV difference is largely driven by high-CNV outliers in the unexposed group with high CN18 (COSMICv3.1[16] copy number signature of unknown etiology), and is not significant ($p = 0.21$) when all samples' spectra are weighted equally to diminish the contribution of outlier samples (the default AMSD method). However, CN9 (diploid chromosomal instability) is higher in professionally exposed patients regardless of how spectra are weighed, so a portion of the signal may result from this contribution as well. This result highlights that AMSD can be applied to any mutation state space, not just SBS in 3mer contexts, and can detect high-mutation outlier biases when samples are weighted by mutation counts.

## Associations between genetic ancestry and tumor mutation spectra

One open question in cancer and evolutionary biology is whether endogenous mutational processes have different biases in populations with different genetic ancestry. Differences in inherited germline mutation spectra have been identified in humans[17,18], and the same genes and exposures that cause these differences might also impact somatic mutagenesis[19]. Exogenous mutational exposures might also show associations with genetic ancestry as a result of differences in environmental variables such as geography and socioeconomic status. Although some ancestry effects on tumor mutagenesis have been identified using mutational signature-based methods[20], to our knowledge there has not been a comprehensive screen for mutation spectrum differences between tumors from different ancestry groups.

The Cancer Genome Atlas (TCGA) is a large compilation of exonic somatic mutations called within thousands of tumors across 33 cancer types[21]. The data set is majority European ancestry (EUR, 83%), but for many cancer types there are sufficient samples of African (AFR, 9.6%) and East Asian (EAS, 6.6%) ancestry to test for ancestry differences in mutation spectrum composition using AMSD[20]. Although this European bias is not ideal for research equity or for power to detect differences, TCGA is still more diverse than other large cancer sequencing cohorts such as PCAWG[22]. We emphasize that associations found within an observational study like TCGA, as opposed to a controlled experimental study like the mouse carcinogen exposures, do not necessarily imply genetic causation given the known association of genetic ancestry with patient sampling biases as well as geographic, socioeconomic, and cultural differences in environmental exposures.

We used AMSD to compare spectra grouped by genetic ancestry within each tumor type for which both ancestries had at least 5 samples. Of 67 comparisons, 16 remained significantly different after Benjamini–Hochberg correction and 6 remained significantly different after Bonferroni correction (Fig. 4A, Supplementary Fig. 7). To interpret what factors may be driving significant differences between ancestries, again we turn to signature analysis, fitting COSMIC v3.4 signatures to individual samples and identifying the largest differences in mean signature exposure (Fig. 4B, Supplementary Fig. 8A). In most cases, SBS1 and/or SBS5, endogenous signatures present across tissues, are the most abundant signatures and are often over-represented in one ancestry group. In some cases, we interpret a higher combined fraction of SBS1 and SBS5 to a lower exogenous signature load, though in other cases the absolute number of SBS1/SBS5 mutations per tumor differs between groups in a way that might be better explained by patient age, as the SBS1/SBS5 "clocklike" mutation load is known to steadily increase over the lifespan (Supplementary Fig. 8B). This second observation indicates that endogenous mutational processes can contribute to ancestry-associated spectrum differences, perhaps due to differences in age at cancer diagnosis, which could correlate with ancestry due to genetic as well as environmental factors.

Lung adenocarcinoma is the only cancer type where we observe significant differences in mutation spectra across multiple population comparisons after Bonferroni correction (AFR vs EAS: $p = 1 \times 10^{-4}$, AFR vs EUR: $p = 6 \times 10^{-4}$). The main driver of this difference appears to be higher AFR exposure to SBS4, a signature closely associated with smoking that mainly consists of C > A mutations (Fig. 4B, Supplementary Fig 9). The difference in SBS4 exposure between AFR and EUR ancestry has been noted before, though notably it has not been attributed to smoking rates, as AFR patients actually appear to smoke significantly less than EUR patients[23]. Among lung adenocarcinoma patients from this study specifically, AFR patients smoked an average of 19.6 pack-years versus 29.8 pack-years in EUR patients ($p = 0.004$, two-sample, two-sided unequal variance $t$ test). This counterintuitive result suggests that this difference may be due to a difference in SBS4 susceptibility, perhaps due to differences in the ability to repair smoke-induced DNA adducts or the immune/inflammatory response to smoking. Some studies have reported individuals of African ancestry to be more susceptible than individuals of European ancestry to the carcinogenic effects of tobacco smoke[24], which might lead to an association

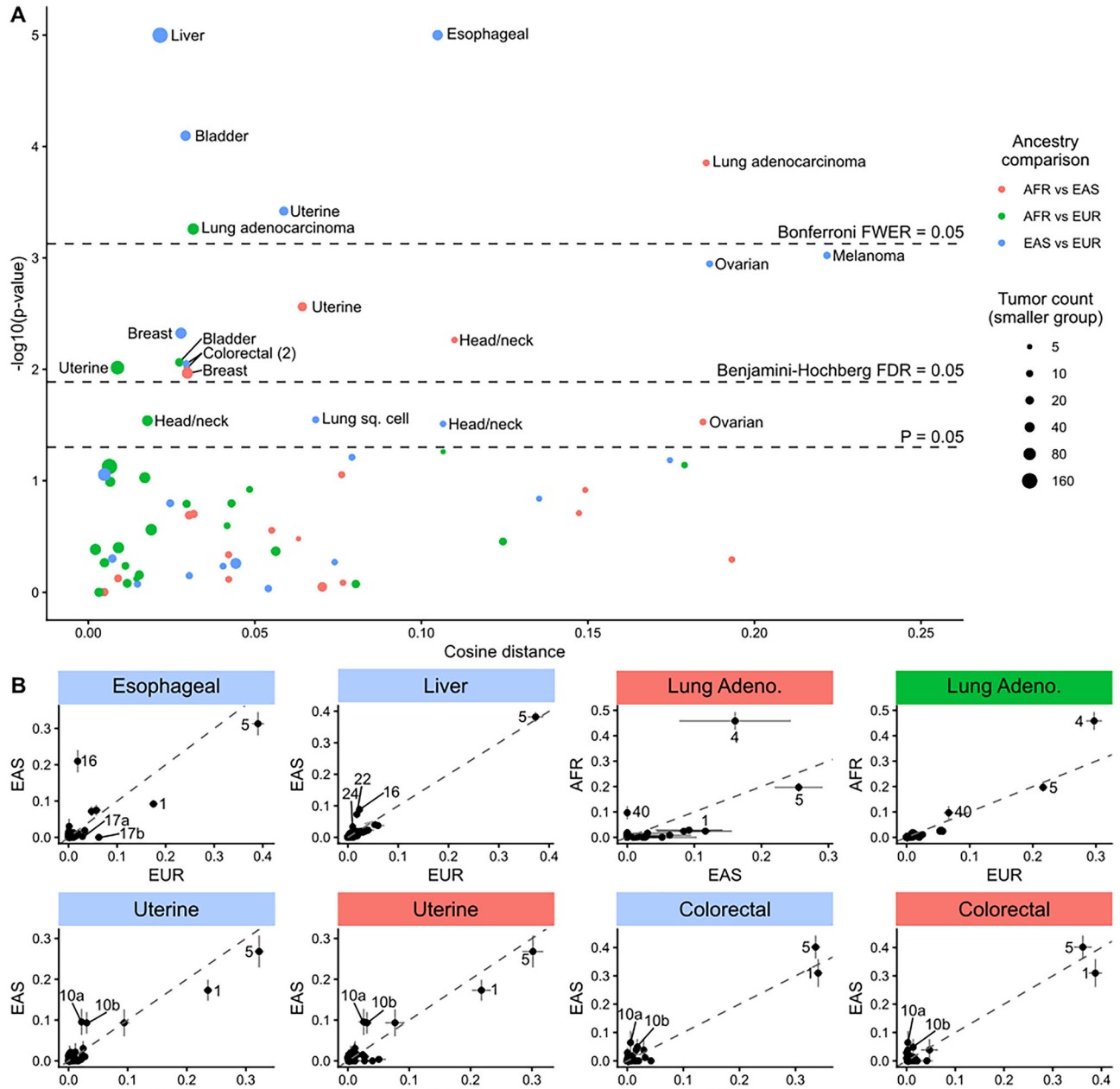

**Fig. 4 | Mutation spectrum differences associated with genetic ancestry.**
**A** Volcano plot of AMSD results comparing TCGA tumors within each cancer type by genetic ancestry. Colors denote pairwise ancestry comparison (AFR vs EAS = red, AFR vs EUR = green, EAS vs EUR = blue), and sizes denote the number of tumors in the smaller of the two groups in each comparison. Dashed lines correspond to significance thresholds: individual $P = 0.05$ threshold, Benjamini–Hochberg

corrected false discovery rate (FDR) adjusted threshold, and Bonferroni corrected family-wise error rate (FWER) adjusted threshold. **B** Mean signature exposure fractions for each ancestry group for eight significantly different comparisons discussed in the text, with each dot denoting a COSMIC v3.4 SBS signature. Diagonal line denotes 1:1 ratio. Error bars denote standard error of the mean.

between SBS4 exposure and African ancestry among lung cancer patients. This finding highlights AMSD's ability to detect a known mutation difference between AFR and EUR lung cancers and extends its generality by also detecting a difference between AFR and EAS lung cancers, despite low EAS sample size ($n = 9$).

Uterine cancer is another tumor type where AMSD detects a difference across multiple ancestry comparisons (EAS vs EUR: $p = 4 \times 10^{-4}$, AFR vs EAS: $p = 3 \times 10^{-3}$, AFR vs EUR: $p = 1 \times 10^{-2}$, all significant at a Benjamini–Hochberg FDR of 0.05). The main driver for the difference between EAS and EUR/AFR appears to be higher EAS exposure to SBS10a and SBS10b, signatures associated with polymerase epsilon exonuclease

domain mutations consisting mainly of TCT > TAT and TCC > TTC mutations (Fig. 4B, Supplementary Fig 10). Interestingly, we observe the same signal in colorectal cancer (EAS vs EUR: $p = 9 \times 10^{-3}$, AFR vs EAS: $p = 1 \times 10^{-2}$, both significant at a Benjamini–Hochberg FDR of 0.05), the other cancer type in which SBS10a/b are regularly detected (Supplementary Fig 10). The increase in SBS10 exposure can be attributed to a higher proportion of tumors with this exposure in EAS for both uterine and colorectal cancer (proportions of uterine cancers with both SBS10a/b are 10/108 AFR (9%), 8/29 EAS (28%), 30/375 EUR (8%), while proportions of colorectal cancers with both SBS10a/b are 3/57 AFR (5%), 3/12 EAS (25%), 11/339 EUR (3%). Previous studies have noted a higher prevalence of POLE

mutations in patients of East Asian ancestry[25–27], but to our knowledge this pattern has not been detected using the prevalence of SBS10a/b.

The two most significant mutation spectrum differences are associated with esophageal cancer and liver cancer, both when comparing EAS to EUR ($p < 1 \times 10^{-5}$, Supplementary Fig 9). Notably, both of these cancer types have been reported to occur at disproportionately high rates in Asian countries[28]. In both cancer types, SBS16 exposure appears higher in EAS. Although this signature's etiology is still unknown according to COSMIC, some studies have linked SBS16 to alcohol consumption[29–31]. Rates of both esophageal cancer incidence and SBS16 exposure covary significantly with geography[32,33], and SBS16 exposure has been linked to genetic variation in aldehyde dehydrogenase 2[29,30], an enzyme involved in alcohol metabolism with a nonfunctional allele common in EAS but rare in other ancestries (rs671)[34–36]. It appears likely that higher SBS16 exposure in EAS is driven in part by greater sensitivity to alcohol exposure despite similar or lower alcohol use, as previously hypothesized regarding SBS4 in AFR[37]. EAS ancestry is also associated with higher exposure to SBS22 in liver cancers. SBS22 exposure has been conclusively attributed to the mutagenic effects of aristolochic acid, a plant compound found in Eastern herbal medicines and contaminated wheat[38].

In esophageal cancers, exposure to SBS17a and SBS17b is elevated in EUR, with both signatures present in 27/125 EUR (30%) versus 0/44 EAS (0%). While SBS17b has been linked to fluorouracil chemotherapy treatment, metadata indicates that fluorouracil was only used to treat a small proportion of esophageal cancer patients and this treatment occurred at similar rates between ancestries (8/126 EUR, 3/44 EAS). If these treatment notes are accurate, then SBS17a/b exposure differences are most likely explained by other putative causes of SBS17a/b, such as reactive oxygen species. While ancestry-related differences have been documented for SBS16 and SBS22, to our knowledge this stark difference in SBS17a/b has not been documented and warrants further investigation.

We observed additional significant mutation spectrum differences in the context of bladder cancer (EAS vs EUR and AFR vs EUR: SBS2/13 - APOBEC cytidine deaminase activity), melanoma (EAS vs EUR: SBS7a/b - ultraviolet light exposure), ovarian cancer (EAS vs EUR: possible sequencing artifacts), head and neck cancer (AFR vs EAS: SBS1/5 - endogenous signatures), and breast cancer (EAS vs AFR and EAS vs EUR: SBS2/13 - APOBEC), summarized in Supplementary Fig 8. Overall, these findings indicate that ancestry-associated mutation biases exist and are highly tumor-type dependent, though we reiterate that they do not necessarily indicate genetic effects, and in many cases may reflect other non-genetic influences that correlate with ancestry. These results also highlight the power of AMSD as a screening tool, with signature analysis interpreting the mutational processes behind significant results.

To investigate whether inferred tumor mutational signature proportions reveal the same association with ancestry that we see when using AMSD to analyze raw mutation spectra, we applied our Wilcoxon rank sum approach to all 67 comparisons (Supplementary Fig 11A). Unlike the mouse carcinogen analysis, where AMSD detected strictly more differences than signature analysis did, we observed several tumor ancestry comparisons where signatures detected a difference that went undetected by AMSD. Upon closer inspection, these outlier results appear to be driven by comparisons with highly uneven sample sizes, with a rare signature exposure in only one or two tumors from the smaller group (Supplementary Fig 11B). For example, a highly significant association of ancestry with the signature profiles of sarcomas (signatures $p = 5 \times 10^{-7}$, AMSD $p = 0.14$) is driven by the presence of artifactual signature SBS54 in 1/6 EAS versus 0/202 EUR, a likely false positive result. In contrast, the ancestry associations detected exclusively by AMSD appear to be more robust, such as the difference between EAS vs AFR lung adenocarcinomas (signatures $p = 0.27$, AMSD $p = 1 \times 10^{-4}$). Although it is possible that other methods for testing for mutational signature profile differences might perform better than our Wilcoxon rank sum method, the lack of a clear counterpart to AMSD in the mutational signature field highlights the value of AMSD as a simple, robust, multiple-testing-minimizing approach to detecting factors that affect mutation spectra.

## Discussion

In this study, we apply the statistical framework of the AMSD permutation test to tumor datasets from mice and humans, revealing that carcinogens and genetic ancestry can affect cancer mutation spectra in situations where group asymmetries previously went undetected using standard signature-based methods. Although conclusively explaining the drivers behind these trends is beyond the scope of this paper, we are able to propose likely drivers through targeted mutational signature analysis of comparisons where AMSD detects a significant asymmetry. These examples demonstrate the power of using AMSD as a hypothesis-testing or screening tool alongside established mutational signature approaches: AMSD directly tests whether mutation spectra differ between groups without requiring prior assumptions about the decomposition of spectra into signatures, while mutational signature analysis reveals complementary information about which biological processes likely contribute to the observed mutation spectrum differences. Used together, these methods offer a more complete picture of how mutational processes vary across biological conditions, enabling both unbiased detection and mechanistic interpretation of differences in genomic integrity.

Although many carcinogens induce very distinctive mutational signatures[8], the majority of carcinogens appear to cause more subtle shifts in mutation spectrum composition, as previously highlighted by Morrison et al. using a metaanalysis of mutational signature dosages[39]. For many carcinogens, AMSD detects subtle differences associated with exposure even when the etiology of these differences remains unresolved. Rather than being attributable to the introduction of new mutational processes, we suspect many of these differences are caused by perturbing endogenous processes via inflammation, immune response, microenvironment, or metabolism/growth rates[40]. Notably, these carcinogens would not necessarily cause a detectable increase in tumor mutation burden if they made small but consistent shifts in endogenous mutational processes. For example, a chemical might cause cancer to develop at a younger age, resulting in a lower proportion of age-associated SBS1 and SBS5 mutations and thus a subtly perturbed average spectrum. Morrison et al. found that carcinogen exposures increase the diversity of mutational signatures within tumors and the homogeneity of signature activity across tumors[39], which may contribute to the mutation spectrum shifts we observe here. These findings underscore the fact that carcinogens can alter mutation spectra through complex and indirect biological pathways, highlighting the value of sensitive, assumption-free methods like AMSD for detecting these patterns.

Despite TCGA's European bias and the limitations of exome-sequencing data, we were able to detect significant mutation spectrum differences between continental ancestry groups across multiple cancer types. While prior work has revealed differences in cancer incidence and mutational burden across populations[20], our results suggest that the underlying mutation processes themselves may also differ in subtle but systematic ways. These differences may be directly mediated by genetic causes of cancer risk, or they might reflect environmental differences that covary with genetic ancestry[41]. Demographic differences between who suffers from cancer in different populations might also play a role, for example if wealthier populations experience later age of cancer onset due to social determinants of health. As larger, more diverse cancer datasets become available, methods like AMSD will be critical for rigorously testing whether mutational processes vary across populations and identifying underlying factors contributing to disparities in cancer risk and outcomes.

Though our main analyses apply AMSD to the most commonly used mutation spectrum state space and data type – SBS 3mer spectra ascertained using bulk tumor sequencing – AMSD is easily applicable to other mutation spectrum state spaces and other data types. This approach is straightforwardly applicable to spectrum data involving double base substitutions, insertion/deletions, structural variants, and copy number alterations (as demonstrated in mesothelioma), as well as extended SBS sequence contexts,

such as 1mer, 5mer, transcribed/untranscribed strand and genic/intergenic contexts. Other sequencing methods, such as single-cell, multi-region, or spatial transcriptomics, provide greater intra-tumor resolution in which AMSD could test for cases when spectra vary across time and space in individual tumors[42]. AMSD can also be applied in a GWAS-like scan for mutator alleles, similar to its initial application in recombinant inbred mice[7], although we caution that preliminary attempts to calculate AMSD-based mutation spectrum associations with millions of variants across the human genome faced a heavy multiple testing burden, so we would recommend using it on target loci or on datasets with more resolution than TCGA. Finally, AMSD is not limited to cancer datasets and could be applied in other capacities, such as germline mutation rate evolution[17] or somatic mutations in healthy tissues[43] when sufficient numbers of mutations or samples are available.

In conclusion, AMSD is a versatile and powerful framework for detecting differences in mutation spectra across diverse biological settings. As sequencing technologies advance and datasets grow larger, tools like AMSD will be essential for identifying subtle shifts in mutational processes that may be overlooked by mutational signature analysis alone. Whether applied to population-level studies, environmental exposure screens or testing individual hypotheses, AMSD provides a statistically grounded method with minimal underlying assumptions that can test for mutation spectrum associations to advance our understanding of mutagenesis and cancer evolution.

## Methods

### The aggregate mutation spectra distance (AMSD) permutation testing pipeline

As input, the AMSD permutation test takes two matrices of mutation spectra, such as a group exposed to a carcinogen and a control group unexposed to a carcinogen. AMSD then aggregates the mutation spectra within each group into a single spectrum, either by taking the average frequency of each mutation type (the default, which weights all samples equally regardless of mutations per sample) or a sum of the count of each mutation type (resulting in the upweighting of samples with higher mutation loads). AMSD then calculates a distance metric (default = cosine distance) between the two groups. To calculate the significance of this distance, AMSD then randomly reshuffles the samples to create the two control groups (same number of samples in each group as the original grouping), computes the aggregate mutation spectra and distance associated with this new grouping, and repeats this reshuffling process for a specified number of permutations (default = 1000) to create a null distance distribution. Then AMSD compares the observed distance to the null distribution, outputting a $p$-value corresponding to the fraction of random reshufflings that produce a distance greater than or equal to the observed distance. This observed versus null comparison can be visualized as a histogram or violin plot, as seen in the paper figures.

### AMSD power analysis with simulated data

In order to test AMSD's ability to detect spectrum differences, we built in a framework to simulate spectra sampled from pre-set mutational signature exposures. The simulation framework randomly samples mutations at probabilities corresponding to their frequencies in supplied signatures (COSMIC v3.2). For the test run in Supplementary Fig. 1A, we set the control group baseline to 30% SBS1, 60% SBS5, and 10% SBS18 to simulate endogenous mutational processes. We then simulate an "exposure" group, in which we sampled from the same baseline distribution of mutational signatures and then sampled additional mutations from one additional signature. We varied four parameters (listed below), running 100 simulations for every combination of these parameters. We then compared the "control" and "exposure" spectra groupings for each simulation using AMSD and for each parameter combination reported the fraction of the 100 simulations for which AMSD detected the difference between the control and exposure group at a significance level of $p < 0.05$.

- Baseline mutations per sample
  - 50 to simulate a whole exome sequencing data set
  - 2500 to simulate whole genome sequencing
- "Exposure" signature added
  - SBS2 to simulate a common "spikey" signature (small number of high-frequency trinucleotides)
  - SBS40 to simulate a common "flat" signatures (relatively equal frequencies of mutation probabilities across many trinucleotide contexts)
- Extra mutations from "exposure" signature, as a % of control mutation count
  - 2% to simulate a weak exposure (e.g. 50 extra mutations for the 2500 baseline mutation group)
  - 5%
  - 10%
  - 20% to simulate a strong exposure
- Number of samples per group
  - 5 to simulate a small experimental study (e.g. Riva et al.)
  - 25
  - 125
  - 625 to simulate large observational study (e.g. breast cancers in TCGA)

### Signature-based statistical test to compare with AMSD

As a reference for how well mutational signature comparisons alone would detect spectra differences, we used a Wilcoxon rank sum test, as previously applied to detect signature differences[9–11]. We used a Wilcoxon rank sum test instead of a $t$ test because normality assumptions were often violated. For each signature present in any sample from a comparison we compared signature exposure fractions by Wilcoxon rank sum test. We then used Bonferroni correction to correct $p$-values for multiple signature tests. Finally, as the overall significance of the difference we take the most significant signature $p$-value.

We applied this approach to test across the simulation parameters listed above, fitting each simulated sample to COSMICv3.4 signatures using SigProfilerAssignment[12] and comparing using the Wilcoxon rank sum method for a difference between the "exposure" and "control" groups. In order to reduce runtime, we did not include the largest sample size comparison (625) and ran 50 simulations per parameter comparison rather than 100. Additionally, we added an additional signature to test the robustness of these methods to the incompleteness of the COSMIC catalog. We chose the signature of exposure to Cyclophosphomide ("CP"), a chemotherapy that causes DNA damage from the COSMIC Experimental Signatures v1.0 (https://cancer.sanger.ac.uk/signatures/experimental/cyclophosphamide/?profile_id=cyclophosphamide_557117b73fe2&variant_class=sbs), since it did not closely resemble any COSMIC signatures. For each parameter set, we reported the fraction of 50 simulations for which each method (AMSD or signatures/Wilcoxon) detected the difference between the control and exposure group at a significance level of $p < 0.05$ (Supplementary Fig. 1B).

### Mouse carcinogen analysis

We downloaded SNV mutation calls from Riva et al.[6] and tallied trinucleotide context mutation counts using Helmsman[44] (v1.5.2) against *M. musculus* (house mouse) genome assembly GRCm38 (mm10). We then used AMSD to compare each carcinogen-exposed tumor type to the control set of spontaneously arising tumors from the same tissue type (liver or lung tumors). We used the default AMSD settings, weighting all sample spectra equally, but with 100,000 random resamplings for the null expectation. Results comparing observed cosine distance to the null distribution are available as violin plots for each AMSD comparison ($n = 29$) in Supplementary Fig. 2. To correct for multiple testing, we used Bonferroni and Benjamini–Hochberg adjusted $p$-values. To show all results on same plot (Fig. 3A), we show a Bonferroni correction threshold ($-\log_{10}(0.05/n)$), and an estimated Benjamini–Hochberg correction threshold from linear

regression of $-\log_{10}$(adjusted $p$-values) against $-\log_{10}$(unadjusted $p$-values) ($R^2 = 0.994$, $\beta = 1.32$) for a threshold of $-\log_{10}(0.05) \times \beta$.

In order to interpret potential mutational signature drivers behind significant results, we also downloaded the mutational signature exposures from Riva et al. In brief, they extracted mutational signatures de novo, compared them to COSMIC signatures to identify similar human signatures, and fit each sample to the mouse mutational signatures, keeping only signature exposures that were at least 5% signature exposure fraction. We compared signature exposures (Fig. 3D, E) using both the Wilcoxon rank sum test and the two-tailed $t$-test with unequal variance. For all signature comparisons involving the highlighted examples of oxazepam and TCP, both tests were in agreement for all signatures (both either $p < 0.05$ or $p > 0.05$). To further investigate what mutation types may contribute to the oxazepam difference, which was not resolved by signatures, we performed a Wilcoxon rank sum test for each of the 96 SBS trinucleotide contexts for oxazapam-exposed liver tumor versus spontaneous liver tumors, with results visualized as a volcano plot in Supplementary Fig 5B. We also re-ran AMSD for this comparison while separating out mutations in genes from mutations in intergenic regions.

We compared the detection ability of AMSD to the Wilcoxon rank sum signature-based approach described above, plotting $p$-values for each comparison in Supplementary Fig 5D. We also compared the mean cosine distance for each comparison to the mean within-sample diversity metric from Morrison et al., and the AMSD -log10($p$-value) to the mean within-sample diversity bootstrap -log10($p$-value)[39]. For each comparison, we grouped by tumor type (lung or liver) and ran linear regression to visualize the relationship, finding results significantly correlated for liver tumors, and nonsignificantly anticorrelated for lung tumors. For mouse liver tumors, mean within-sample diversity and its significance correlated with cosine distance (Supplementary Fig. 5E) and AMSD $p$-values (Supplementary Fig. 5F) respectively, but was nonsignificantly anticorrelated for lung tumors. This indicates that this metric is highly dependent on the signatures present in the reference spontaneous tumors, and though results may overlap since they are both driven by spectra differences, this metric is ultimately testing for a different signal than AMSD (variability vs distance).

### Asbestos exposure analysis

We downloaded SNV, CNV, and SV mutation calls from Mangiante et al.[15] and tallied these respective mutation counts in standard COSMIC format using Helmsman[44] (v1.5.2) for SNVs or SigProfilerMatrixGeneratorR[45] (v1.2) for CNVs and SVs. For each mutation type, we grouped tumors by whether the patients had been professionally exposed to asbestos or not professionally exposed to asbestos and ran AMSD with 10,000 random resamplings for the null expectation. We ran both AMSD methods for each - weighting all sample spectra equally with aggregate mutation spectra means and weighting samples by mutation count with aggregate spectra sums. We also downloaded copy number mutational signature exposures from Mangiante et al., calculating the mean of each signature exposure fraction (for "means" - weighted evenly), count (for "sums" weighted by mutation counts), or count after dropping outlier samples with >500 copy number variants ("sums, no outliers") for the exposed/unexposed groupings. We also calculated the standard error of the mean ($\sigma/\sqrt{n}$) for each mean to estimate uncertainty, plotted in Supplementary Fig. 6G–I.

### TCGA ancestry analysis

We downloaded TCGA SNV mutation calls from Ellrott et al.[21] and tallied trinucleotide context mutation counts using Helmsman[44] (v1.5.2) against the human genome assembly GRCh37 (hg19). We then grouped tumors by consensus superpopulation ancestry calls from Carrot-Zhang et al.[20], removing patients that were not majority African (AFR), East Asian (EAS) or European (EUR) ancestry due to low South Asian and Admixed American sample sizes. We also removed tumors with <10 mutations so that these samples would not be evenly weighted with tumors with more mutation type resolution. We then ran pairwise comparisons using AMSD

for all cancer types for which both ancestries in the comparison had at least 5 samples. We performed analysis based on the major TCGA cancer type classifications (https://gdc.cancer.gov/resources-tcga-users/tcga-code-tables/tcga-study-abbreviations), but used the colloquial naming in the text and figures. For the full list of comparisons ($n = 67$), see results in Supplementary Fig. 7, with tumor types following standard TCGA abbreviations. We used the default AMSD settings, weighting all sample spectra equally, but with 100,000 random resamplings for the null expectation. To correct for multiple testing, we used Bonferroni and Benjamini–Hochberg adjusted $p$-values. To show all results on same plot (Fig. 4A), we show a Bonferroni correction threshold ($-\log_{10}(0.05/n)$), and an estimated Benjamini–Hochberg correction threshold from linear regression of $-\log_{10}$(adjusted $p$-values) against $-\log_{10}$(unadjusted $p$-values) ($R^2 = 0.999$, $\beta = 1.45$) for a threshold of $-\log_{10}(0.05) \times \beta$.

In order to identify which mutational signatures were likely to be driving significant AMSD differences, we fit each individual sample to the COSMIC SBS v3.4 mutational signature database using SigProfilerAssignment[12] using default parameters and exome = TRUE. We then plotted mean exposures across samples, grouped by ancestry and tumor type, as shown in Fig. 4B and Supplementary Fig. 8, to identify which outlier signatures were overrepresented on one ancestry for significant AMSD differences. Since we used the AMSD implementation with all samples weighted evenly, we show mean exposure fractions for our main results, but provide mean absolute exposure mutation counts for further context in Supplementary Fig. 8B. Finally, we compared the detection ability of AMSD to the Wilcoxon rank sum signature-based approach described above, plotting $p$-values for each comparison in Supplementary Fig. 11A. To investigate outlier comparisons that were significant by one method but not the other, we plotted the Wilcoxon rank sum results for each signature as a volcano plot in Supplementary Fig. 11B to identify the driver signature(s), and plotted a histogram of exposures for that signature.

### Mutation-weighted versus unweighted AMSD results

The default AMSD implementation uses mutation spectrum fractions as an input, weighing all samples equally regardless of the number of mutations. In order to test how this implementation compared to a mutation-weighted approach, using absolute mutation counts rather than fractions, we ran each for the mouse carcinogen comparisons and the TCGA ancestry comparisons, plotting the -log10($p$-value) outputs in Supplementary Fig. 12. For the mouse carcinogen comparison, results were largely comparable, with equal weighting having slightly more power to detect significant differences than mutational weighting. Given this is a well-controlled experiment under laboratory conditions, it is not surprising that outliers did not drastically affect results. Equal weighting was much more sensitive at detecting differences in the TCGA ancestry comparisons, with differing results highlighting how drastically these two methods may differ in real-world messier data sets, where outliers are likely to be more common, for real or artifactual reasons. In general, we recommend using the default equal-weighting approach for detecting consistent differences between groups, but note that mutation-weighting comparisons may be appropriate for testing whether outlier samples are over-represented in one group, as seen in the asbestos example.

### Reporting summary

Further information on research design is available in the Nature Portfolio Reporting Summary linked to this article.

### Data availability

All analyses in this study use publicly available datasets, and figures and results can be reproduced using the code available at https://github.com/sfhart33/AMSD_cancer_mutation_spectra. Preprocessed mutation spectra are included in the repository, while raw data can be accessed from:
- Mouse carcinogen exposure: https://github.com/team113sanger/mouse-mutatation-signatures/blob/master/starting_data/snvs.rds

**Article**

- Asbestos exposure: https://github.com/IARCbioinfo/MESOMICS_data/tree/main/phenotypic_map/MESOMICS
- TCGA ancestry metadata: https://gdc.cancer.gov/about-data/publications/CCG-AIM-2020
- TCGA somatic mutations: https://gdc.cancer.gov/about-data/publications/mc3-2017

The original implementation of the AMSD as a method for identifying mutator alleles is also available on github: https://github.com/quinlan-lab/proj-mutator-mapping.

## Code availability

The Aggregate Mutation Spectrum Distance permutation test is implemented as the R package "mutspecdist", available at https://github.com/sfhart33/mutspecdist. All analyses in this study use publicly available datasets, and figures and results can be reproduced using the code available at https://github.com/sfhart33/AMSD_cancer_mutation_spectra.

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

## Acknowledgements

We thank Harris and Feder lab members for figure feedback, Sayre Coombs for graphic design feedback, and Tom Sasani for developing the original implementation of AMSD and manuscript feedback. This work was possible due to funding from NIH training grant T32-HG000035 supporting S.F.M.H., Worldwide Cancer Research grant 24-0106 to N.A., NIH grant 1DP2CA280623-01 to A.F.F., NIH NIGMS grant 2R35M133428-06 to K.H., and the Allen Discovery Center for Cell Lineage Tracing. Where authors are identified as personnel of the International Agency for Research on Cancer/World Health Organization, the authors alone are responsible for the views expressed in this article and they do not necessarily represent the decisions, policy or views of the International Agency for Research on Cancer/World Health Organization. The results published here are in part based on data generated by the the TCGA Research Network (https://www.cancer.gov/tcga) and by the Rare Cancers Genomics initiative (www.rarecancersgenomics.com).

## Author contributions

S.F.M.H., A.F.F., and K.H. contributed to study conceptualization and design. S.F.M.H. performed the data analysis. S.F.M.H., N.A., A.F.F., and K.H. interpreted the results. S.F.M.H. wrote the original draft of the manuscript. S.F.M.H., N.A., A.F.F., and K.H. contributed to review and editing of the manuscript.

## Competing interests

The authors declare no competing interests.
