## [Transparent Peer Review file · Communications Biology]

A signature-agnostic test for differences between tumor mutation spectra reveals carcinogen and ancestry effects

Corresponding Author: Dr Kelley Harris

This manuscript has been previously reviewed at another journal. This document only contains information relating to versions considered at Communications Biology.

Version 0:

Reviewer comments:

Reviewer #1

(Remarks to the Author)

The authors have provided substantial improvements to the manuscript, including: 1) Clarifications about the utility of AMSD and its added value compared to traditional signature decomposition methods.

2) Confirmation of the superiority of AMSD over signature extraction in specific problems that signature analysis is limited in solving on its own. 3) The inclusion of a Wilcoxon rank-sum approach for mutational signature composition (applied across simulated, murine, and TCGA analyses), offering a more direct comparison to AMSD. As a result of these thorough revisions, the authors have successfully addressed my concerns.

Response to manuscript reviews

We thank our two anonymous reviewers for their helpful comments and suggestions. In response, we have reshaped the paper in three major ways that we believe have strengthened and clarified the narrative:

- 1. We now provide evidence for a key point we failed to communicate clearly: that mutational signature decomposition always introduces statistical uncertainty and error, so when a hypothesis can be tested using raw spectra alone, this is intrinsically a more reliable option.**
- 2. Related to point #1, we have attempted to compare AMSD's performance to the performance of a similar method that compares mutational signature decomposition profiles rather than raw mutation spectra. The results highlight AMSD's usefulness as a screening tool and show that raw mutation spectra can contain patterns that become obscured by signature inference, especially in certain situations where signature inference is most error-prone.**
- 3. In our real data analysis sections, we now explicitly discuss which of our findings confirm published results (which provide important methodological validation) and which results we believe to be novel.**

Please see our point-by-point response below in bold.

Reviewer #1, expertise in cancer genomics, mutagenesis and mutational analyses (Remarks to the Author):

In this manuscript, Hart et al present the aggregate mutation spectrum distance (AMSD) as a metric to identify differences in the mutation spectrum between groups. They propose AMSD as a first step to determine significant differences in mutagenic profiles before more in-depth interpretation using mutational signatures, and provide evidence suggesting that AMSD may be able to detect differences that were previously overlooked by conventional signature analysis. This approach is put to the test in two examples: 1) a study of carcinogen exposure in mice, and 2) TCGA data comparing the mutation profiles across ancestry groups.

The manuscript presents compelling evidence that AMSD can be valuable when conventional signature analysis faces methodological constraints (e.g., low mutation count). However, the added value of AMSD when signature analysis is feasible remains less clear. My main concerns are the following:

Major comments:

- The analysis in the simulated datasets assessing AMSD's ability to detect mutation spectrum differences is insightful, as it shows this tool performs well in situations in which signature analysis might be limited. However, it's not fully clear how AMSD compares to conventional mutational signature analysis in these scenarios in identifying differences in the mutation

spectra. Could you clarify whether conventional signature analysis would be able to successfully extract the underlying signatures in these situations?

To address this question, we created a framework for directly comparing AMSD to an approach that is rooted in mutational signature analysis. Since we know of no published method for testing whether two groups of tumors have significantly different mutational signature compositions, we created our own test that uses signature exposures rather than raw spectra as its input but is otherwise similar to AMSD. Specifically, we chose to compare signature exposures for each signature present in the comparison by Wilcoxon rank sum test, then Bonferroni-correct for the number of signatures tested in that comparison. We acknowledge that other approaches could be used to compare the signature decomposition profiles of two groups (involving different choices of statistical tests, signature fitting methods, fitting parameters). However, we also note that any alternative method would involve bespoke implementation and that AMSD avoids the need to make many of these methodological choices:

“In theory, the uncertainty associated with mutational signature decomposition guarantees that testing for a difference between the raw mutation spectra of two groups should be less noisy than testing for a difference in signature composition. To empirically test whether this is the case, we compared AMSD’s performance to a Wilcoxon rank sum test approach that several studies have used to test individual signatures for differences between groups^{9–11}. We extended this Wilcoxon rank-sum test approach to test for differences in overall mutational signature composition and applied it to our simulated data as follows: first, we identified exposures of COSMIC signatures within each simulated sample using SigProfilerAssignment¹². We then used the Wilcoxon rank sum to test each individual signature for an exposure difference between groups, applied a Bonferroni multiple testing correction, and identified the most significant corrected signature p-value. This method performed similarly to AMSD across many simulations, but severely underperformed AMSD in cases with i) small sample size, ii) flat signatures, iii) spiky signatures at low mutation counts, and iv) non-COSMIC signature exposure (Supp Fig 1B). The results of this test suggest that performing signature decomposition as a preprocessing step does not decrease uncertainty in high-information settings where signature decomposition is very reliable, but that running tests on raw mutation spectra is better practice when data is sparse or signature inference is error-prone for other reasons. It is possible that other tests for differences in signature composition might perform better, as we did not explore the performance of other signature fitting tools, parameters, or significance methods, each which would incur inherent trade-offs and multiple testing burdens across signatures. It is also worth noting that fewer such choices are necessary when running AMSD to screen for differences in raw mutation spectra, a fact that contributes to our tool’s robustness.”

- In situations where mutational signatures can be reliably extracted (e.g., sufficient mutation count and samples per group), the precise added value of employing AMSD prior to signature analysis is not as clear.

We have rewritten the introduction to emphasize that there is not an established signature-based method for answering the particular question that AMSD focuses on: whether two groups of tumors have significantly different mutation profiles. It may be straightforward to perform signature analysis on the two groups of samples and observe that signature exposures are distributed unevenly within and between groups, but some other downstream test is then required to decide whether variation between the groups exceeds the variation that exists within the groups. Figure 1 now illustrates some of the complexities that are associated with this downstream statistical task. For example, if mutational signature analysis reveals that the median members of two groups are largely similar but that both groups contain different numbers of outliers whose mutation spectra are divergent in various ways, this decomposition does not render any judgement about whether the groups are significantly different from one another. Part of what the AMSD does is quantify whether the two groups have significantly different distributions of outlier spectra. This is important in the setting of cancer exposures since many carcinogens likely act heterogeneously among individuals, and we want to be able to evaluate whether a carcinogen appears to produce extreme signatures even a small proportion of the time.

A second point we now emphasize is that the reliability of mutational signature deconvolution is never absolute. When mutation counts and sample sizes are large, reliability gets better, but ultimately mutational signatures and their dosages are hidden variables that must be inferred from data using a model, which always leaves room for error and the impact of model assumptions. Since mutational signature deconvolutions do not come with error bars, it is not obvious when looking at them whether they are extremely stable extractions from large amounts of data or are based on bare minimum numbers of mutations. In contrast, raw mutation spectra are observed sample features that can be objectively compared (modulo the impact of mutation calling errors, which also affect downstream signature inference).

Although there is no obvious established tool for doing what AMSD does within a mutational signature-based framework, we were able to construct such a method by drawing inspiration from Wilcoxon rank sum analyses that several previous studies used to compare the prevalence of a single signature between groups. We now apply this alternative signature-based approach to the mouse carcinogen and TCGA ancestry section to compare its performance to AMSD's in the context of simulations as well as real-data examples, including those with substantial prior knowledge of mutational signatures (e.g. TCGA). We find that AMSD generally functions better than signature approaches for performing an unbiased screen of many comparisons. We have also added further discussion throughout the paper to clarify this point.

- Regarding the analysis of carcinogens in the murine models, the authors mention “qualitative differences among the proportions of signature exposures in several other carcinogen-exposed

tumors from this dataset". This statement suggests that conventional signature analysis (as performed in Riva et al.) would be able to identify different fractions of endogenous signatures related to some of the exposures. If this is the case, what would be the added value of AMSD in this situation? The current manuscript only shows two examples, and only one presents no significant differences in the signature profile across groups. Expanding this analysis in other examples would help clarify this issue.

We agree that the mutational signature deconvolutions performed by Riva, et al. reveal some of the same signal that AMSD does, in the sense that signature exposure distributions appear different between different carcinogen-exposed groups. However, a downstream statistical method is still needed to decide whether these differences between groups are significant or not. We now fill this gap using our Wilcoxon rank sum test method and we compare the results of this signature-based analysis to our AMSD results (see results in Supplementary Figures 3, 4, and 5). Ultimately our signature-based test was not able to detect the signals that AMSD detected due to low sample sizes and increased multiple testing burden per comparison for multiple signatures:

"To directly compare a signature-based approach to signature-agnostic AMSD, we applied the Wilcoxon rank-sum approach described in the previous section to all 29 comparisons (Supp Fig 5A). AMSD was much more powerful at detecting differences, with the signature-based approach weakened by the combination of multiple testing burden (2-7 signatures for each of 29 comparisons) and low sample sizes ($n \leq 12$)."

- In the analysis of mutation spectra across ancestry groups, signature analysis is performed on the pooled mutations of each sample grouping. Could this approach sufficiently solve the issues of low mutation counts and low sample sizes, potentially reducing the need for AMSD? The results indeed suggest that significant differences in signature profiles can be found using this approach despite low sample size.

This mutation pooling approach falls short in that it erases variation among the tumors present within each ancestry group. This within-group variation is taken into account by AMSD to determine whether between-group variation is significant. As in our benchmarking analysis and our mouse carcinogen analysis, we now use our Wilcoxon rank-sum approach to analyze the TCGA mutational signature composition in a way that is more directly comparable to AMSD. Specifically, we now fit signatures to each individual sample, rather than to the aggregate spectra. One point of confusion may be that we have error bars on these plots that reflect fitting error, rather than standard deviations. To clarify this point we altered our approach to fit individual samples, which captures the inter-sample variability and largely leads to the same signature signals as our previous pooled fitting approach.

In some cases, our Wilcoxon rank sum analysis of signature exposures misses differences that AMSD detects, while in other cases the signature approach detects a

difference that does not hold up as robust under scrutiny. We summarize these results in the following paragraph:

“To investigate whether inferred tumor mutational signature proportions reveal the same association with ancestry that we see when using AMSD to analyze raw mutation spectra, we applied our Wilcoxon rank-sum approach to all 67 comparisons (Supp Fig 11A). Unlike the mouse carcinogen analysis, where AMSD detected strictly more differences than signature analysis did, we observed several tumor ancestry comparisons where signatures detected a difference that went undetected by AMSD. Upon closer inspection, these outlier results appear to be driven by comparisons with highly uneven sample sizes, with a rare signature exposure in only one or two tumors from the smaller group (Supp Fig 11B). For example, a highly significant association of ancestry with the signature profiles of sarcomas (signatures $p = 5e^{-7}$, AMSD $p = 0.14$) is driven by the presence of artifactual signature SBS54 in 1/6 EAS versus 0/202 EUR, a likely false positive result. In contrast, the ancestry associations detected exclusively by AMSD appear to be more robust, such as the difference between EAS vs AFR lung adenocarcinomas (signatures $p = 0.27$, AMSD $p = 1e^{-4}$). Although it is possible that other methods for testing for mutational signature profile differences might perform better than our Wilcoxon rank-sum method,, the lack of a clear counterpart to AMSD in the mutational signature field highlights the value of AMSD as a simple, robust, multiple-testing-minimizing approach to detecting factors that affect mutation spectra.”

Please note that this new analysis utilizes SigProfiler to extract mutational signatures, whereas our original draft utilized the method sigfit instead. We made this change because SigProfiler outperforms sigfit at fitting individual samples with low mutation counts. This resulted in some differences in signature exposures, most notably more SBS5, which SigProfiler incorporates as a prior since it is expected to be present ubiquitously across tissues, and low fitting values rounded to zero. This discrepancy highlights how the need to choose a signature fitting method introduces further uncertainty beyond what is captured by the signature-based p -values in our manuscript.

- Comparisons of the mutation spectra across groups are based on signature fractions. However, changes in signature burdens (i.e., absolute values) also provide relevant information about the mutagenic potential of exposures, which is overlooked if only focusing on fractions. Furthermore, signature fractions are inherently affected by the presence of other signatures. The mutational insults captured by different signatures are often independent and therefore should be analysed as such. In this regard:

1. Would analysing the signature burdens in absolute values eliminate the reported artefactual differences in SBS1 fraction across groups?
2. Are there significant differences in absolute SBS4 burden in LUAD across ancestry groups? Is it possible that the lower relative SBS4 fraction in EUR patients is due to the presence of other signatures in this group?

We added a comparison of absolute signature mutation burdens to our analysis (Supp Fig 8B), but kept the main interpretation focus on signature fractions to reduce the impact of high mutation load outliers, and because we applied AMSD using equally weighted samples rather than weighted by mutation count. This resolved the observation of increased SBS1 (now SBS1&5 - see previous response) in some cases, but not all cases. As for SBS4, this signature still appears to drive the difference when considering the absolute burden. We edited the following paragraph to incorporate these new results:

“To interpret what factors may be driving significant differences between ancestries, again we turn to signature analysis, fitting COSMIC v3.4 signatures to individual samples and identifying the largest differences in mean signature exposure (Figure 4B, Supp Fig 8A). In most cases, SBS1 and/or SBS5, endogenous signatures present across tissues, are the most abundant signatures and are often over-represented in one ancestry group. In some cases, we interpret a higher combined fraction of SBS1 and SBS5 to a lower exogenous signature load, though in other cases the absolute number of SBS1/SBS5 mutations per tumor differs between groups in a way that might be better explained by patient age, as the SBS1/SBS5 “clocklike” mutation load is known to steadily increase over the lifespan (Supp Fig 8B). This second observation indicates that endogenous mutational processes can contribute to ancestry-associated spectrum differences, perhaps due to differences in age at cancer diagnosis, which could correlate with ancestry due to genetic as well as environmental factors.”

- The authors state that their approach reveals “that carcinogens and genetic ancestry can affect cancer mutation spectra in situations that previously went undetected using standard signature-based methods”. However, some of the signature differences across ancestry populations have been previously described, e.g. SBS4 in EAC in EUR vs AFR and SBS16 in alcohol-related cancers in EAS vs other ancestries. In this sense, could authors elaborate on the added value of their proposed approach?

We have now clarified which of our observations confirm previous reports, which have not been previously reported based on signature analysis but do have some other kind of prior evidence, and which observations are novel to the best of our knowledge. For example, we edited the following relevant sentence in the abstract, in addition to relevant results section:

“Some of the asymmetries driving these differences have been previously reported, such as elevated SBS4 in African lung adenocarcinomas and SBS16 in East Asian esophageal/liver cancers, and some have not been previously reported, such as elevated SBS17a/b in European esophageal cancers and SBS10a/b in East Asian uterine/colorectal cancers.”

Minor comments:

- As stated in the manuscript, comparison of ancestry groups present many limitations, especially regarding the fact that the analysis does not control for geographic, socioeconomic, or cultural differences in environmental exposures. Given this, I would advise greater caution in concluding that genetic ancestry can affect cancer mutation spectra.

We agree that we do not want the reader to come away with the wrong idea about what can be concluded about ancestry-related mutation biases. We added the following sentences to introduce and conclude the ancestry results section:

“Exogenous mutational exposures might also show associations with genetic ancestry as a result of differences in environmental variables such as geography and socioeconomic status.”

“We reiterate that these biases do not necessarily indicate genetic effects, and in many cases may reflect other non-genetic influences that correlate with ancestry.”

Before presenting any TCGA results, we lead with the following disclaimer:

“Of note, when interpreting these results from an observational study like TCGA, as opposed to a controlled experimental study like the mouse carcinogen exposures, significant differences from AMSD imply an association, not necessarily genetic causation. In addition to genetic effects caused by allele frequency differences between ancestries, tumor mutation spectra might also be impacted by patient sampling biases or geographic, socioeconomic, or cultural differences in environmental exposures.”

...and the way we revisit the topic in the discussion:

“These differences may be directly mediated by genetic causes of cancer risk, or they might reflect environmental differences that covary with genetic ancestry. Demographic differences between who suffers from cancer in different populations might also play a role, for example if wealthier populations experience later age of cancer onset.”

...that we are properly cautioning the reader about how to interpret these results.

Reviewer #1 (Remarks on code availability):

The results of the paper seem reproducible based on the code provided. Furthermore, the code is well annotated and provide detailed installation information. Therefore, it could be a usable resource for the community.

We appreciate this feedback and we hope that the quality of the code base will make AMSD more widely applicable in practice than mutation spectrum comparison methods that are primarily implemented for use in a single study.

Reviewer #3, expertise in cancer evolution, cancer genomics and mutagenesis (Remarks to the Author):

In Hart et al, the authors apply AMSD - a permutation test to assess differences in mutation spectra - to cancer genomes. They show that it is a powerful tool for identifying significant differences in mutational spectra. In particular, they apply AMSD to mouse genomes exposed to carcinogens, where it identifies mutation spectrum shifts associated with carcinogen exposures, and to TCGA, where it highlights ancestry-related differences in mutation spectra across multiple cancer types.

The core concept of the paper, a simple method to identify shifts in mutation spectra, is both timely and could be of significant use within the expanding field of mutational signatures. The manuscript is well presented and methodologically sound.

However, for a broad audience, I unfortunately found the work to be underdeveloped, both methodologically and biologically. Methodologically, when this method should be preferred in place of others isn't fully addressed. Biologically, there was an absence of investigation into causes of the spectrum shifts - without this, most biological results were confirmation of prior findings or as expected from orthogonal info (e.g. chemo used less in a given patient population ~ less of chemo associated signature). The method has also been previously published and the findings based on mouse experimental data have considerable similarities with the preprint of Morrisson et al (<https://doi.org/10.1101/2023.11.23.23298821>). More detailed comments and suggestions are given below.

We appreciate the positive feedback on the rigor of the manuscript and the method's potential utility. To improve the motivation for the utility of the method, we have reworked the paper's introduction to avoid attempting to compare AMSD to mutational signature analysis writ large, which was a poor choice in retrospect given that signature analysis is used for so many different things whereas AMSD is useful for a more specific problem that signature analysis does not adequately solve on its own. We now clarify that AMSD is a powerful tool for quantifying whether there is a significant difference in mutation spectrum composition between two heterogeneous groups of tumors, and our new Figure 1 is designed to illustrate the methodological gap between performing signature analysis and adequately answering this question.

Major comments:

1) When should this method be preferred, or not, as compared to other methods? I suggest that the simulation analyses should be compared when using signature exposures, the hypergeometric test in Jiang et al (<https://doi.org/10.7554/eLife.68285>), the method in Kucab et

al, and the method in Morrisson et al (if the authors have a strong reason against including one of these, then that's fine but just write why they're excluding it in the text). While I like the simplicity of AMSD the authors haven't demonstrated when the new method improves upon prior work.

This comment helped motivate us to better describe the methodological niche that AMSD fills, which is not the same niche that is filled by any of the other named methods. We have added a further discussion of these other methods to the introduction, but we did not explicitly compare these methods with AMSD side-by-side due to the following differences between their objectives and limitations:

- 1. A hypergeometric test allows you to compare a single sample to another single sample, but if it is being used to compare two groups of signals, it does not account for variation within either group. In particular, if two groups of samples are identical except for the fact that one group contains an extreme outlier, the hypergeometric test would average the outlier into its group's mean spectrum, potentially leading to the detection of a significant difference between groups when AMSD would likely not identify a significant difference.**
- 2. Of the approaches named by the reviewer, the one developed by Kucab et al. is most directly comparable to AMSD. However, it requires setting an arbitrary threshold for determining which comparisons are significant (top 10% of differences in their paper), which is based on the specific structure of their data rather than a general null distribution. Though this works for their study, this setup would make it difficult to apply the method to other datasets without extensive customization. Relatedly, this method is not engineered for easy applicability to other datasets, but would likely need to be re-implemented by other investigators seeking to use it.**
- 3. Morrison et al. pose a different question and test for within-sample signature diversity rather than signature/spectra differences between groups.**

Although none of these three methods is directly comparable to AMSD, we appreciate the need to compare AMSD to a signature-based alternative given that many papers prefer to infer mutational signatures as a preprocessing step rather than analyzing mutation spectra directly. To meet this need, we developed our own Wilcoxon rank sum-based test for signature composition differences, and we now compare this test to AMSD within each major section of the paper (see responses to Reviewer #1).

2) If differences in spectrum exist according to AMSD but not signature exposures, the next steps to determine aetiology should be further investigated in at least 1 case. It seems this can be done with the data in hand.

E.g. taking the case of oxazepam, does the significance hold with both weighted/unweighted AMSD? If we take the ratio of the spectra (exposed/spontaneous) which mutation classes are enriched/depleted? How does the comparison in Figure 2d vary if the exposures are not

normalised to sum to 1? Using AMSD does the difference persist if restricting to genic vs intergenic regions or early/late replication timing or some other genomic feature?

Such an analysis would show that once AMSD demonstrates a difference in spectrum, there's further analyses which more finely determine how mutagenesis is altered, as opposed to the course-grained signature exposures. E.g. perhaps the mutation burden is broadly decreased in genic regions due to transcriptional changes induced by the carcinogen. Currently, if AMSD finds a difference, in the absence of signatures, it's not clear how to proceed. It may well be that no further explanatory factors are found, but a more detailed attempt should be made.

We have deepened our analysis of the oxazepam example to provide a framework for how to follow up on results that are not resolved using signatures. We now show that oxazepam's mutagenic effects appear to be most pronounced in intergenic regions and in certain trinucleotide contexts that are not well captured by signatures from the available catalog. This likely implies that either 1) the current mutational signature catalog does not capture the source of the oxazepam example and/or 2) more samples may be needed to resolve the signature or signatures that drive the effect.

“For oxazepam, differences in signature exposure fractions are not significantly different for any individual signature (Figure 3C,E), but signature decomposition appears to be masking subtle but consistent mutation spectrum differences that exist between oxazepam-exposed and unexposed tumors, such as ACT>ATT mutations (Figure 3B, Supp Fig 5B). This spectrum difference was somewhat more pronounced in intergenic regions ($p = 0.003$, mean mutations/sample = 1166) than genic regions ($p = 0.04$ mean mutations/sample = 618). It is possible this difference could be explained by the presence of an unknown signature, or that instead of introducing new mutational processes oxazepam might be shifting tumor cell growth/metabolism or DNA repair in ways that perturb the relative rates of endogenous mutational processes. Ultimately, a larger sample set and complete signature set would be needed to resolve what mutational processes may be driving this difference, but this analysis demonstrates how AMSD and signature fitting can work in concert.”

(Note that Figure 2 is now Figure 3)

3) Given that Morrison et al have also analysed the data of Riva et al, and also found that many of the carcinogens induce changes in “mutational signature patterns”, I believe a careful comparison between the findings of this paper and those of Morrison et al is warranted, potentially as a supplementary note. E.g. they also find that oxazepam alters mutation patterns.

We have added a side-by-side comparison of the mouse carcinogen AMSD results with the same comparisons Morrison et al. and discussed the results. Interestingly, there is a strong relationship between their metric and ours for liver cancers, but not for lung

cancers. Ultimately, the Morrison, et al. metric is measuring something slightly different from what is measured by AMSD, but the metrics are correlated since they are both mutation spectrum based and influenced by variability within and between groups of samples.

“We also compared the mean cosine distance for each comparison to the mean within-sample diversity metric from Morrison et al., and the AMSD $-\log_{10}(p\text{-value})$ to the mean within-sample diversity bootstrap $-\log_{10}(p\text{-value})$ ³⁹. For each comparison, we grouped by tumor type (lung or liver) and ran linear regression to visualize the relationship, finding results significantly correlated for liver tumors, and nonsignificantly anticorrelated for lung tumors. For mouse liver tumors, mean within-sample diversity and its significance correlated with cosine distance (Supp Fig 5C) and AMSD p-values (Supp Fig 5D) respectively, but was nonsignificantly anticorrelated for lung tumors. This indicates that this metric is highly dependent on the signatures present in the reference spontaneous tumors, and though results may overlap since they are both driven by spectra differences, this metric is ultimately testing for a different signal than AMSD (variability vs distance).”

Minor comments:

-Throughout it's entirely unclear whether the weighted or unweighted AMSD is used. Particularly important in the TCGA analysis, e.g., hypermutant tumors might contribute disproportionately. I'd like to see both for all the biological findings.

We now clarify in the introduction that samples are weighted equally in all analyses except for the asbestos analysis, where we highlight how outliers can have different effects depending on sample weighting. In addition, we now run both weighted and unweighted approaches for the mouse and TCGA comparisons, compare the findings in Supp Fig 12, and discuss the discrepancies in the paper methods:

“The default AMSD implementation uses mutation spectrum fractions as an input, weighing all samples equally regardless of the number of mutations. In order to test how this implementation compared to a mutation-weighted approach, using absolute mutation counts rather than fractions, we ran each for the mouse carcinogen comparisons and the TCGA ancestry comparisons, plotting the $-\log_{10}(p\text{-value})$ outputs in Supp Fig 12. For the mouse carcinogen comparison, results were largely comparable, with equal weighting slightly outperforming mutation-weighting at detecting significant differences. Given this is a well controlled experiment under laboratory conditions, it is not surprising that outliers did not drastically affect results. Equal weighting was much more sensitive at detecting differences in the TCGA ancestry comparisons, with differing results highlighting how drastically these two methods may differ in real-world messier data sets, where outliers are likely to be more common, for real or artifactual reasons. In general, we recommend using the default equal-weighting approach

for detecting consistent differences between groups, but note that mutation-weighting comparisons may be appropriate for testing whether outlier samples are over-represented in one group, as seen in the asbestos example.”

-Not clear what's novel in the asbestos analysis, more comparison should be made with findings of original study.

We clarified the findings of the original paper to highlight the novelty of our findings:

“Though they did not find asbestos-related mutational signatures or spectrum changes, Mangiante et al. detected molecular differences that correlate with asbestos exposure, such as the expression of mesenchymal and neoangiogenesis-related genes¹⁶, which could indirectly influence mutation spectra in subtle ways that signature analysis may not be powered to detect.”

-How much of the ancestry findings would be made using the differences in exposures?

We have added this analysis - see responses above to reviewer #1.

-Please cite prior work regarding sbs22/24 being enriched in asian populations, e.g. Steve Rozen's work.

We have added citations to these prior studies, along with clarification of which ancestry results recapitulate previous findings (such as this observation), versus which we believe to be novel.

-Given that experimental signatures of 5-Fu treatment exist and the proportions of treated patients are known, it would be interesting to know how much the differences in SBS17b can be explained by 5-Fu.

We have elaborated on this analysis as an example of a novel finding that does not appear to be explained by differences in 5-Fu treatment, adding the following paragraph:

“In esophageal cancers, exposure to SBS17a and SBS17b is elevated in EUR, with both signatures present in 27/125 EUR (30%) versus 0/44 EAS (0%). While SBS17b has been linked to fluorouracil chemotherapy treatment, metadata indicates that fluorouracil was only used to treat a small proportion of esophageal cancer patients and this treatment occurred at similar rates between ancestries (8/126 EUR, 3/44 EAS). If these treatment notes are accurate, then SBS17a/b exposure differences are most likely explained by other putative causes of SBS17a/b, such as reactive oxygen species. While ancestry-related differences have been documented for SBS16 and SBS22, to our knowledge this stark difference in SBS17a/b has not been documented and warrants further investigation.”

-I was left wondering if different cancer subtypes had been aggregated together, and whether there is systemic difference in subtype frequency in different ancestry groups, e.g. are all liver cancers HCC? A Supplementary csv file to go with Fig 3a would address this or at least state in text.

We have added clarification to the text (Note that Figure 3 is now Figure 4):

“We performed analysis based on the major TCGA cancer type classifications (<https://gdc.cancer.gov/resources-tcga-users/tcga-code-tables/tcga-study-abbreviations>), but used the colloquial naming in the text and figures.”

-Line 365 “Notably, these carcinogens would not necessarily cause a ...” is important but very difficult to parse.

We have edited this paragraph to clarify our thinking on this issue, specifically adding a clearer example: “a chemical might cause cancer to develop at a younger age, resulting in a lower proportion of age-associated mutations and thus a subtly perturbed average spectrum”

-In Supplementary Figure 1, legend says Extra mutations (%) but (I think) the numbers in the legend are fractions (0.02-0.2, whilst in the caption it refers to 2-20%)

We have corrected this error.

-I don't think Table 1 adds much, these aren't problems AMSD solves.

We have replaced Table 1 with a new figure, “Problems with distinguishing whether two groups of samples have different mutational signature exposures”, that focuses more narrowly on describing the group comparison problem that AMSD is designed to solve.

-Suggest cite the published version of Sasani et al.

We have updated this citation.

Reviewer #4, ECR (Remarks to the Author):

Reviewer #4 (Remarks on code availability):

REVIEWERS' COMMENTS:

Author responses in bold

Reviewer #1 (Remarks to the Author):

The authors have provided substantial improvements to the manuscript, including: 1) Clarifications about the utility of AMSD and its added value compared to traditional signature decomposition methods.

2) Confirmation of the superiority of AMSD over signature extraction in specific problems that signature analysis is limited in solving on its own. 3) The inclusion of a Wilcoxon rank-sum approach for mutational signature composition (applied across simulated, murine, and TCGA analyses), offering a more direct comparison to AMSD. As a result of these thorough revisions, the authors have successfully addressed my concerns.

Thank you Reviewer #1 for taking the time to review our revised manuscript and comment on the other reviewers concerns.

Reviewer #1 (Comments on the authors' responses to Reviewers #2/3):

Many of the concerns of Reviewers #2/3 overlap with my comments in the previous round of revisions, which have now been addressed (e.g., unclear utility of AMSD and added value of the biological results). Furthermore, the authors have effectively responded to Reviewers #2/3's request for a comparison of AMSD with other existing methods (hypergeometric test, and methods proposed by Kucab et al and Morrison et al). This information has been incorporated into the discussion and in supplementary figure 5.

Regarding results from the mouse model, the authors have further investigated the mutational spectrum in a case with significant differences according to AMSD (i.e., oxazepam exposure), providing an example of how this tool can work in conjunction with signature analysis. However, the following minor points should be addressed:

- The current manuscript states that spectrum differences between samples exposed vs unexposed to oxazepam were "somewhat more pronounced" in intergenic regions compared to genic regions. To further support this claim, the mutation spectra per trinucleotide context across genomic regions should be provided.

Thank you for this useful suggestion. We have now added the genetic and intergenic mutation spectra for oxazepam-exposed and spontaneous tumors to Supplementary Figure 5C. With this closer look we realized we cannot conclude that the difference is more pronounced given intergenic has more mutations and the cosine distances are similar in between genic and intergenic comparisons, so we changed this sentence to:

“This spectrum difference was present in both intergenic regions ($p=0.003$, mean mutations/sample = 1166) and genic regions ($p=0.04$ mean mutations/sample = 618) (Supp Fig 5C).”

- To provide further details about the similarities and differences of the mutation profile in mice from the different groups in Figure 3c, the signature burden per sample should also be provided in absolute burdens.

Thanks also for suggesting this figure’s inclusion. We have now included a plot of absolute mutation burden by attributed signatures in Supplementary Figure 5A to match the samples shown in Figure 3C.